# Lifecycle of a predatory bacterium vampirizing its prey through the cell envelope and S-layer

Yoann G. Santin [1,4], Adrià Sogues [2,3,4], Yvann Bourigault[1], Han K. Remaut [2,3] & Géraldine Laloux [1] ✉

Predatory bacteria feed upon other bacteria in various environments. *Bdellovibrio exovorus* is an obligate epibiotic predator that attaches on the prey cell surface, where it grows and proliferates. Although the mechanisms allowing feeding through the prey cell envelope are unknown, it has been proposed that the prey's proteinaceous S-layer may act as a defensive structure against predation. Here, we use time-lapse and cryo-electron microscopy to image the lifecycle of *B. exovorus* feeding on *Caulobacter crescentus*. We show that *B. exovorus* proliferates by non-binary division, primarily generating three daughter cells. Moreover, the predator feeds on *C. crescentus* regardless of the presence of an S-layer, challenging its assumed protective role against predators. Finally, we show that apparently secure junctions are established between prey and predator outer membranes.

Predatory bacteria are ubiquitously found across a wide array of environmental niches, where they play a pivotal role in shaping bacterial communities. Their antagonistic interactions involve distinct killing strategies that predators employ to ensure their proliferation[1]. For instance, the opportunistic predator *Myxococcus xanthus* utilizes a highly coordinated social motility strategy, resembling a "wolf-pack" behavior, combined with the transient deployment of a contact-dependent killing apparatus, to efficiently prey upon other bacteria[2,3]. Obligate predatory bacteria exclusively proliferate in a prey-dependent manner. They do so as endo- or epibiotic predators that feed on the prey's cellular content by growing and proliferating within the periplasmic space of diderm bacteria[4-6], or while attached to the prey cell surface[7-11], respectively, through mechanisms that remain largely mysterious.

In response to predation pressure, bacteria have evolved a range of adaptive behaviors that serve as defense mechanisms, impeding predator assaults or hindering predator proliferation within populations. Various mechanisms have been extensively studied in resistance against protozoa, nematodes[12], and bacteriophages[13]. However, research on defenses against obligate predatory bacteria has been comparatively limited. To date, effective prey resistance mechanisms against the endobiotic predator *Bdellovibrio bacteriovorus* have not been discovered[14]. Despite the identification of a transcriptional "scream" response in prey upon *B. bacteriovorus* predation[15], the functional significance of the induced genes remains uncertain, as these responses might signify the prey's unsuccessful attempts to maintain homeostasis. So far, the surface layer (S-layer) represents the only reported defensive structure against obligate predatory bacteria[16,17], based on previous observations including that *Caulobacter crescentus* strains carrying an S-layer were not preyed upon by the obligate epibiotic predator *Bdellovibrio exovorus*[10,18]. Notably, these studies chiefly led to the more general notion that protection against predatory bacteria is one of the roles of the S-layer. The S-layer is a paracrystalline protein monolayer that surrounds almost all Archaea and many bacterial species[16,19,20]. Recent investigations used *C. crescentus* as a model organism to characterize the in situ structure of the complete S-layer at a near-atomic resolution[21,22]. Using electron tomography and integrated structural analyses, these studies revealed that in Gram-negative bacteria, the S-layer is anchored to the outer membrane via the lipopolysaccharide O-antigen and is stabilized by

[1]de Duve Institute, UCLouvain, 75 avenue Hippocrate, 1200 Brussels, Belgium. [2]Structural and Molecular Microbiology, Structural Biology Research Center, VIB, Pleinlaan 2, 1050 Brussels, Belgium. [3]Structural Biology Brussels, Vrije Universiteit Brussel, Pleinlaan 2, 1050 Brussels, Belgium. [4]These authors contributed equally: Yoann G. Santin, Adrià Sogues. ✉e-mail: geraldine.laloux@uclouvain.be

multiple $Ca^{2+}$ ion bonds, forming a porous yet highly stable structure. The S-layer of *C. crescentus* is composed of the single self-assembling protein RsaA, accounting up to 30% of the total protein synthesis[23]. Despite the considerable energy investment required for its assembly, the precise function of this S-layer remains elusive[16]. While the S-layer has been described to stabilize the bacterial envelope[24], act as a permeability barrier, or function as an adhesion factor, its specific biological role is unidentified in numerous species[16]. The mechanism by which the S-layer could prevent predatory attacks remains ambiguous.

Epibiotic predation is proposed to be a common behavior in the environment, which might precede endobiotic predation[25]. While our comprehension of endobiotic predation has deepened over the last years largely from studies using *Bdellovibrio bacteriovorus* as a model organism[26], the lifecycle and predation mechanisms of obligate epibiotic predators, such as the closely related species *B. exovorus*[10], remain enigmatic. This is mostly due to the current lack of live imaging capturing their predatory behavior and proliferation. Notably, the dynamics of prey consumption with respect to predator growth and division are unclear. *B. exovorus* was proposed to proliferate through binary division[10,11,27,28], but this is solely supported by electron microscopy images of a few predivisional *B. exovorus* attached to *C. crescentus* cells. Besides the α-proteobacterium *C. crescentus*[10,11,27,28], only a handful of species, belonging to distinct families in the gamma- and beta-proteobacteria classes, were reported to serve as *B. exovorus* prey, mostly with weaker prey lysis capacities[10,11,27]. This is unlike the relatively broad prey range identified for *B. bacteriovorus*; however, the extent of the *B. exovorus* prey range remains to be explored further.

Here, we set out to provide the first view of the epibiotic lifecycle of *B. exovorus* by monitoring the prey-predator interaction, the predator growth and division cycle, as well as the fate of the prey cell, through time-lapse imaging and cryo-electron microscopy. Moreover, we provide new insights into the range of bacterial species that are susceptible to *B. exovorus* attacks. Imaging and quantifying predation on isogenic prey strains also allowed us to revisit the impact of the S-layer on epibiotic predation. Altogether, this study reveals the novel proliferation mode of this predator on the surface of its prey and hints at the establishment of tight contact between prey and predator outer membranes that challenges the proposed defensive role of the S-layer against predatory bacteria.

## Results

### *B. exovorus* vampirizes a restricted prey range

To get insights into the predatory lifestyle of *B. exovorus*, we cultivated the reference strain JSS in the presence of an S-layer deficient mutant (ΔrsaA) of the *Caulobacter crescentus* reference strain NA1000 (also known as CB15N) and imaged cells by phase contrast microscopy. Cell ghosts featuring the typical *C. crescentus* morphology and numerous smaller vibrioid *B. exovorus* cells were visible after overnight co-incubation of both species, indicating predation (Fig. 1a). Beside *C. crescentus*, no other α-proteobacterium was described as prey for *B. exovorus*. Here we challenged a set of species belonging to several families of the α-proteobacteria class (5 Caulobacteraceae: *Asticcaulis excentricus, A. biprosthecum, A. benevestitus, Brevundimonas subvibrioides, Phenylobacterium lituiforme*; 2 Rhizobiaceae: *Agrobacterium tumefaciens, Sinorhizobium meliloti;* 1 Brucellacea: *Ochrobactrum anthropi*; 1 Hyphomonadacea: *Hyphomonas neptunium*) (Fig. 1b), and *E. coli* as negative control[10] with *B. exovorus*. We did not observe an increase in predator or prey ghost numbers for *E. coli* and the tested Hyphomicrobiales (i.e., Rhizobiaceae and Brucellaceae) (Fig. 1c, black species names), indicating no predation on these species. However, *B. exovorus* consumed and proliferated on all tested

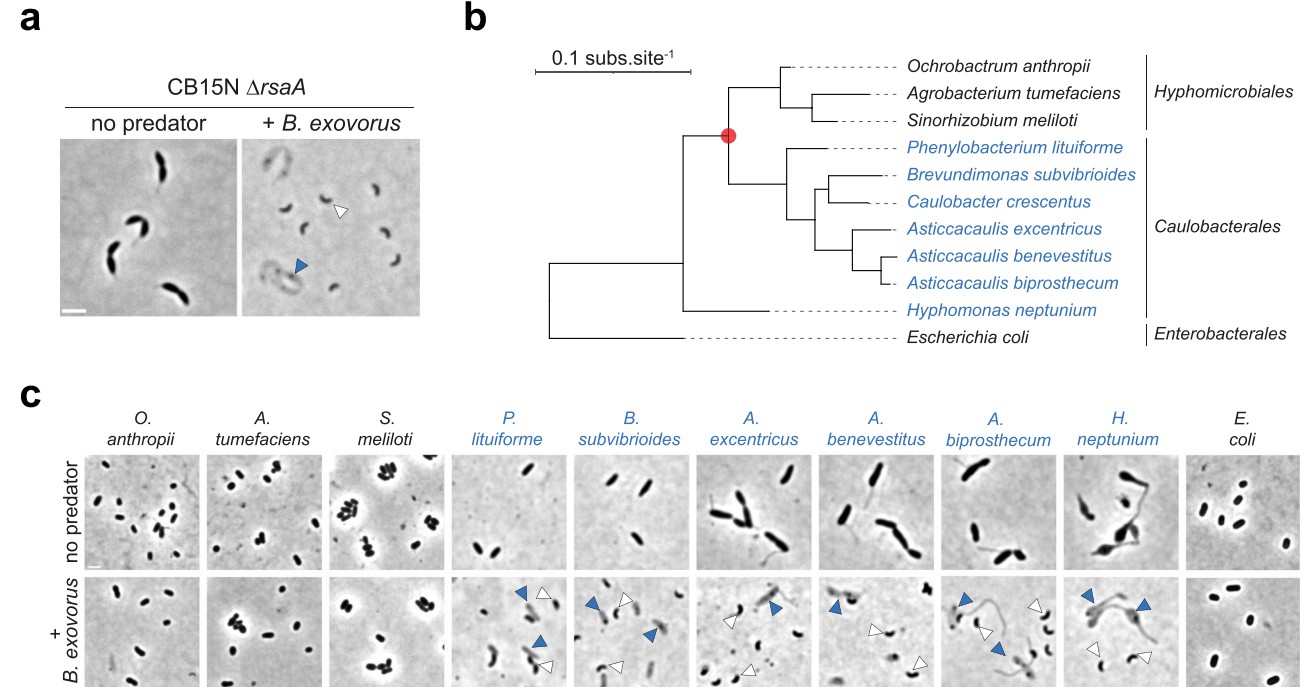

**Fig. 1 | *B. exovorus* predation of α-proteobacterial species is not limited to *C. crescentus*. a** Representative phase contrast images of an overnight mix between the S-layer deficient *C. crescentus* CB15N ΔrsaA mutant and *B. exovorus*. *C. crescentus* ghost cells (blue arrowhead) and high numbers of smaller vibrioid cells (*B. exovorus*, white arrowhead) are only observed upon mixing with *B. exovorus*. Scale bar, 2 μm. **b** Phylogeny of the selected α-proteobacterial species was derived from 16S rRNA sequences (using *E. coli* as an outgroup). Species susceptible to predation by *B. exovorus* are designated in blue. The red circle represents the evolutionary event separating Hyphomicrobiales and Caulobacterales. **c** Representative phase contrast images of overnight mixes between *B. exovorus* and the selected α-proteobacterial preys or *E. coli* as a negative control. Positive predation is evidenced by the presence of ghost cells (blue arrowheads) and the proliferation of *B. exovorus* predators (white arrowheads). Scale bar, 2 μm.

Caulobacterales (i.e., Caulobacteraceae and Hypomonadacea) species (Fig. 1c, blue species names). Taken together, these findings suggest that (i) this epibiotic predator feeds upon a limited prey range within α-proteobacteria that is not restricted to *C. crescentus* and (ii) the presence or absence of specific factors within the Hyphomicrobiales order confers resistance to *B. exovorus* predation.

## The S-layer does not protect *C. crescentus* from predation by *B. exovorus*

Previous reports using *B. exovorus* as a predator[10,18] led to the generally established idea that the S-layer prevents bacterial predation[16,17]. We, therefore, sought to examine how this protein monolayer, which surrounds wild-type *C. crescentus*[21,22], impacts predation by *B. exovorus*. Surprisingly, the reference strain CB15N, which expresses the *rsaA* gene forming a functional S-layer[29], was similarly predated as the S-layer-deficient CB15N Δ*rsaA* mutant strain (Fig. 2a compared to Fig. 1a). Kinetics measurements of prey lysis showed that there was no striking difference in predation efficiency at the population level between the two *C. crescentus* strains (Fig. 2b and Supplementary Fig. 1a, b). Consistently, the quantification of predator attachment to both prey strains showed no statistical delay in prey binding ($4.1 \pm 4.4\%$ difference in the fraction of predators attached to wild-type vs Δ*rsaA C. crescentus* upon 15 min post-infection; Fig. 2c). Additionally, *B. exovorus* cell size parameters were undistinguishable upon predation on wild-type or Δ*rsaA C. crescentus* (Supplementary Fig. 1c). Finally, cryo-electron microscopy imaging showed that all *C. crescentus* cells to which a *B. exovorus* cell was attached are surrounded by a continuous S-layer (Fig. 2d and see below). Hence, our data demonstrate that *B. exovorus* can predate upon *C. crescentus* carrying an S-layer, challenging the proposed role of the S-layer in protecting against predatory bacteria.

## *B. exovorus* uses a novel non-binary division pattern producing triplet progenies

To obtain a more detailed view of the entire *B. exovorus* life cycle, we flowed predator cells in a microfluidics chamber at the bottom of which an unsynchronized population of *C. crescentus* cells were immobilized (see Methods) and performed time-lapse imaging by phase contrast microscopy. *B. exovorus* swam at high speed, attached to *C. crescentus* within seconds (Supplementary Movie 1), and elongated over time (Fig. 3a and Supplementary Movie 2). Predators were positioned on the prey surface without an apparent bias towards a specific *C. crescentus* cell type (swarmer vs stalked) or cell body area (Supplementary Fig. 1d). The *C. crescentus* stalk, which is a thin extension of the cell including all envelope layers[30], also appeared to serve as a contact site on which *B. exovorus* could attach and grow (Supplementary Fig. 1d, right, Supplementary Fig. 1e, and Supplementary Movie 3). Invasion of the prey cell by the predator was never observed, confirming previous reports that *B. exovorus* displays an epibiotic predatory lifestyle[10,11].

The predivisional stage of the *B. exovorus* lifecycle could be visualized in phase contrast by the constriction of their cell body. Intriguingly, predivisional *B. exovorus* cells frequently displayed more than one (typically two) constriction sites, indicating that these predators rely on a non-binary mode of cell division (Fig. 3a, magenta arrowheads). Quantification from these time-lapse experiments revealed that the majority of *B. exovorus* cells produced three daughter cells ($76 \pm 3.3\%$, $n = 413$, Fig. 3b), while two progenies were observed in a smaller fraction of the population ($24 \pm 3.2\%$, $n = 413$, Supplementary Fig. 2a, cyan arrowhead). Using a relatively short time interval of 2 min, we determined that the release of *B. exovorus* progenies is sequential, where the outermost cell and the prey-attached cell leave first and last, respectively (Fig. 3a and see below). These time-lapse experiments also revealed that *B. exovorus* initiates cell constriction during growth (Fig. 3a, b and Supplementary Movie 2), suggesting an intricate

coordination of cell wall synthesis processes responsible for cell elongation and division.

*B. exovorus* cell cycle progression was also captured by cryo-electron microscopy (cryo-EM, Fig. 3c), which confirmed the presence of two constriction sites on elongating predator cells, and their sequential division (Fig. 3c). This intriguing division pattern produces healthy progeny and is unlikely to result in the generation of minicells, as DAPI staining indicated that all future *B. exovorus* daughter cells within predivisional triplets contained DNA (Supplementary Fig. 2c). Newborn cells featured a well-defined nucleoid (Supplementary Fig. 2d) reminiscent of the compact nucleoid reported in *B. bacteriovorus*[31]. Importantly, the timing of *B. exovorus* growth and cell division was similar when wild-type or Δ*rsaA C. crescentus* was used as prey (Supplementary Fig. 2e, f), demonstrating that the S-layer does not impair these key steps of *B. exovorus* epibiotic proliferation, even at the single-cell level. Altogether, these data highlight that *B. exovorus* cell division is not strictly binary and thereby more complex than initially considered[10,11].

## The number of *B. exovorus* progenies is not determined by prey cell size

The closely related endobiotic predator *B. bacteriovorus*, which grows inside its prey bacterium, uses non-binary division to produce odd or even progeny numbers, with reports of 2 to 17 daughter cells released per generation[32–34]. We recently established that *B. bacteriovorus* progeny numbers are determined by the size of the prey cell in which they grow[32], which led us to assess if the apparent restricted variability in *B. exovorus* daughter cell numbers is explained by the relatively fixed *C. crescentus* cell size. Strikingly, comparable distributions of offspring numbers were measured (Fig. 3b) upon time-lapse imaging of *B. exovorus* predation on the elongated *popZ*::Ω *C. crescentus* mutant strain[35] (Fig. 3d). Moreover, while four progenies were also observed (Fig. 3b and Supplementary Fig. 2f, g), it only occurred in rare instances (4.9%, $n = 201$ cells). These findings hint that the predator offspring is not dictated by prey size. Thus, *B. exovorus* and *B. bacteriovorus* use distinct mechanisms to control their division cycle and regulate their progeny numbers.

## The prey cytoplasmic content is digested in situ by *B. exovorus*

Our time-lapse phase contrast microscopy data (Fig. 3a) show that the prey is fading out over time while *B. exovorus* is attached to its cell surface, consistent with the idea that the predator digests the prey cell to fuel its growth and replication. To get insights into the fate of the prey cell during predation, we first labeled the cytoplasm of wild-type *C. crescentus* with the freely diffusing mCherry fluorescent protein (*C. crescentus*[mCh]) and monitored fluorescence intensity during the prey-predator interaction (Fig. 4a and Supplementary Fig. 3). As anticipated, mCherry fluorescence intensity decreased over time in the *B. exovorus*-attached prey compared to uninfected *C. crescentus* cells, indicating digestion of the proteinaceous cytoplasmic content (Fig. 4a and Supplementary Fig. 3). Importantly, there was no transfer of fluorescence signal from *C. crescentus*[mCh] to *B. exovorus* (Fig. 4a and Supplementary Fig. 3), strongly suggesting that digestion of the prey proteins takes place in situ before uptake into the predator cell. The cytoplasmic fluorescence signal completely disappeared at a time point that shortly precedes the detachment of the last predator from the prey, or exactly at that timepoint. Thus, the leak of remaining proteins from the prey and predator detachment can occur within a short (<2 min) time-window. These data imply that *B. exovorus* does not consume the entire proteome of the prey, and that the release of the last attached *B. exovorus* leaves an open wound through which the leftover *C. crescentus* cytoplasmic content is released.

Labeling of the *C. crescentus*[mCh] prey DNA at several time points after mixing with *B. exovorus* showed that the nucleoid is also altered

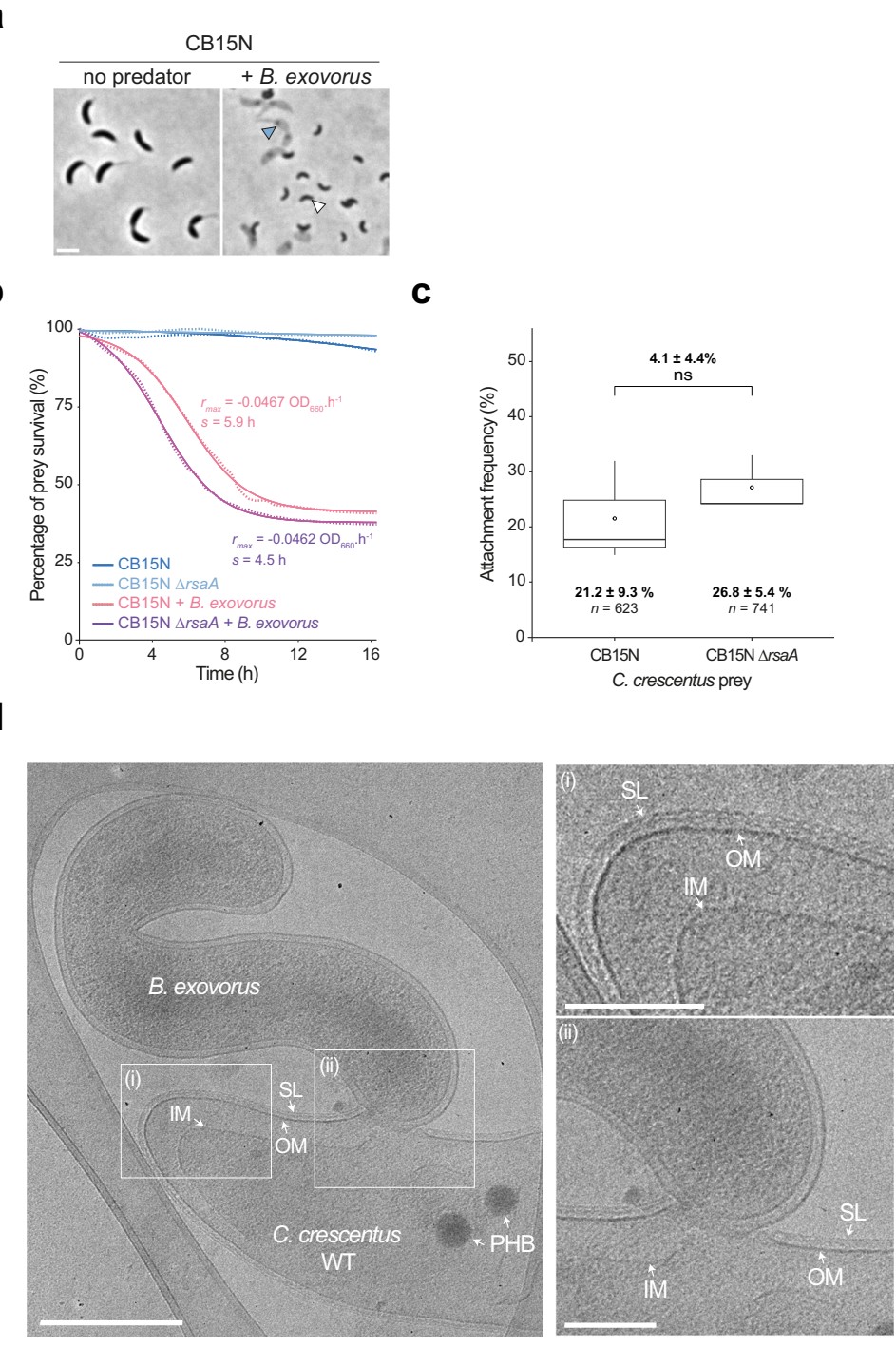

**Fig. 2 | The S-layer does not prevent *B. exovorus* predation. a** Representative phase contrast images of an overnight co-culture between the wild-type *C. crescentus* CB15N and *B. exovorus*. *C. crescentus* ghost cells (blue arrowhead) and newborn *B. exovorus* predators (white arrowhead) are shown. Scale bar, 2 μm. **b** Similar antibacterial efficiency of the *B. exovorus* predator was measured when the wild-type or the Δ*rsaA C. crescentus* strains are used as a prey. *C. crescentus* prey cells were co-incubated with *B. exovorus* in a microplate for 16 h at 30 °C. Optical density at 660 nm (here represented as the percentage of the initial population) was monitored over time, and metrics were extracted using CuRveR[41]. $r_{max}$ corresponds to the killing rate, and *s* is the time point at which $r_{max}$ reaches its maximum value. Colored lines correspond to the fit, and dots are values obtained from three technical replicates. The assay was performed four times, and a representative result is shown. **c** Boxplot representation of the frequency of attachment events of *B. exovorus* onto the wild-type or Δ*rsaA C. crescentus* strains 15 min after flowing the *B. exovorus* cells into the microfluidics chamber. Values correspond to the fraction of prey cells in contact with a predator cell. Bold horizontal bars represent the median value; empty circles represent the mean; the lower and upper boundaries of the internal box plot correspond to the 25th and 75th percentiles, respectively; the whiskers represent the 10th and 90th percentiles. Mean values ± standard deviation and the number of analyzed cells (*n*) are indicated below. Pairwise comparison and the value of the mean difference and the standard deviation from three biological replicates are indicated above the plot (ns nonsignificant, *p* = 0.4; two-sided two-sample Fisher–Pitman permutation test). Source data are provided as a Source Data file. **d** Representative cryo-EM images of *B. exovorus* attached to the wild-type (WT) *C. crescentus* CB15N cell surface (left). Scale bar, 0.5 μm. Magnifications of the selected regions representing the prey envelope layers (top right), and the tight contact between the predator and the prey outer layers (bottom right). PHB polyhydroxybutyrate granule, SL S-layer, OM outer membrane, IM inner membrane. The experiment was repeated twice with similar results. Scale bar, 0.2 μm.

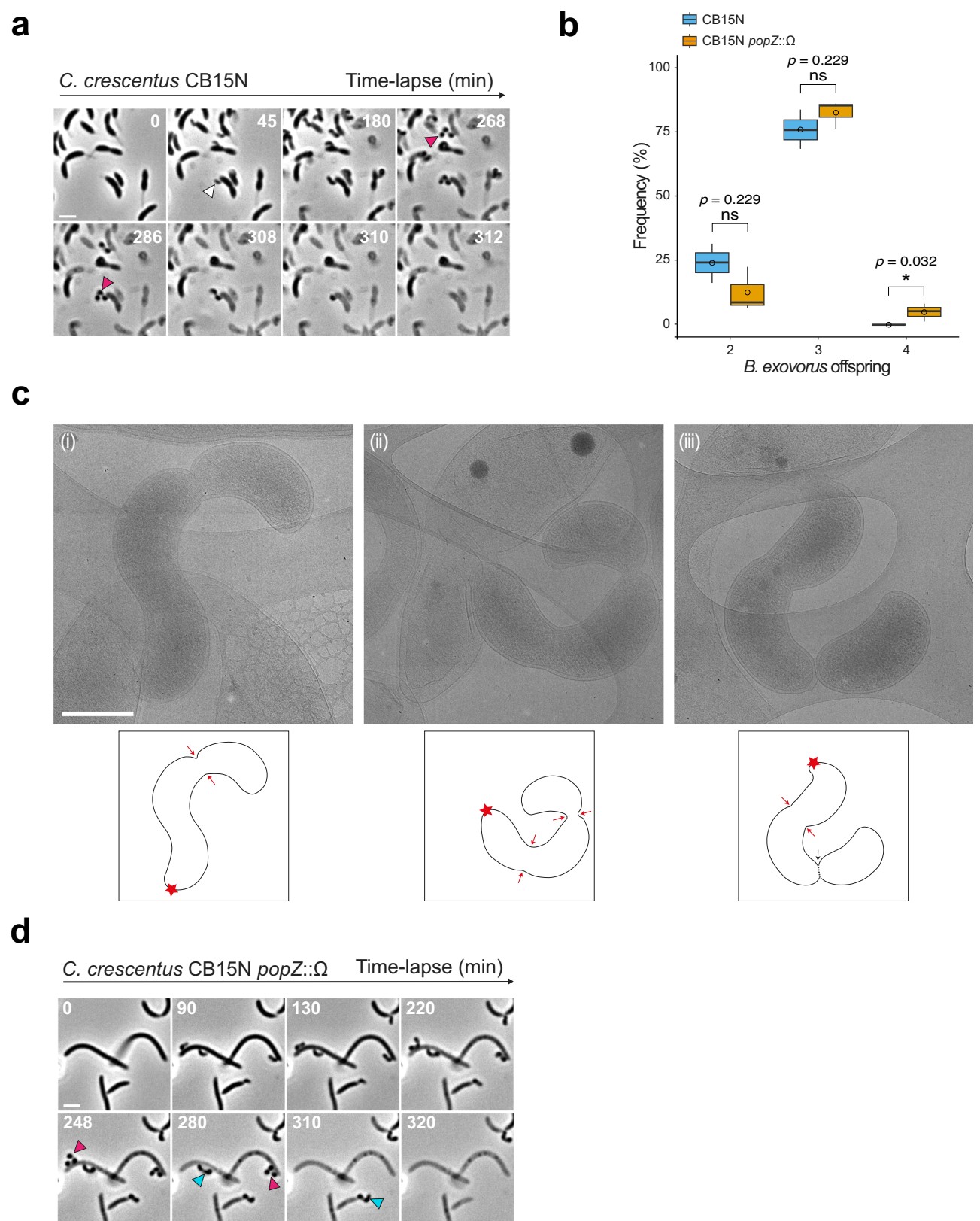

during predation (Fig. 4b). In contrast to uninfected *C. crescentus*, DAPI fluorescence signal intensity rapidly decreased during predation imaged in time-course (Fig. 4b). This observation indicates in situ digestion of the DNA material by *B. exovorus*, which appeared faster than the digestion of the proteinaceous content when compared with the decay of fluorescence intensity from cytosolic mCherry (Fig. 4a, b and Supplementary Fig. 3) or the nucleoid-associated protein HU fused to YFP (HU1-YFP) imaged in the same cells (Fig. 4c). Similarly, cryo-EM imaging of the early and late stage feeding events show a gradual loss in granularity in the prey cellular content (representing proteins, ribosomes, and chromosomal DNA) (Fig. 4d).

**Fig. 3 | *B. exovorus* displays a novel division pattern regardless of prey size.** *B. exovorus* mainly produces triplet progenies. **a** Representative time-lapse phase contrast microscopy images of *B. exovorus* growing onto the wild-type *C. crescentus* prey. The white arrowhead indicates the attachment of a *B. exovorus* cell to a *C. crescentus* cell. Magenta arrowheads show the production of three progenies. Scale bar, 2 μm. **b** Boxplot representation of the frequency of 2, 3, or 4 *B. exovorus* progenies using either the wild-type (blue; *n* = 413) or *popZ*::Ω (orange; *n* = 201) *C. crescentus* strains as prey. Quantification based on Fig. 3a, d. Bold horizontal bars represent the median value; empty circles represent the mean; the lower and upper boundaries of the internal box plot correspond to the 25th and 75th percentiles, respectively; the whiskers represent the 10th and 90th percentiles. Pairwise comparisons from three biological replicates are indicated above the plots (ns nonsignificant; *\*p* < 0.05, two-sided Wilcoxon's *t*-test). Source data are provided as a Source Data file. **c** Representative cryo-EM images of *B. exovorus* growing onto the wild-type *C. crescentus* prey. Each image corresponds to one late step of the *B. exovorus* growth, including the formation of the first constriction site at the distal end of the filament (i), the formation of the second constriction site (ii), and the sequential progenies release (iii). Scale bar, 0.5 μm. The experiment was repeated twice with similar results. Hand-drawn schematic representations based on the cryo-EM image are shown below. Red star corresponds to the predator–prey contact site. Red and black arrows indicate *B. exovorus* constriction and division sites, respectively. **d** Representative time-lapse phase contrast microscopy of *B. exovorus* growing onto the *C. crescentus popZ*::Ω mutant strain as a prey. Cyan and magenta arrowheads show the production of 2 or 3 progenies, respectively. Scale bar, 2 μm.

## Outer and inner prey envelope layers follow distinct fates during predation by *B. exovorus*

Whereas the *C. crescentus* cytoplasm was emptied upon departure of the last *B. exovorus* (Fig. 4a and Supplementary Fig. 3), the prey ghosts retained their typical crescent shape, reminiscent of sacculi (Fig. 4e). Consistently, labeling of the prey peptidoglycan (PG) with the fluorescent D-amino acid HADA[36] prior to the addition of *B. exovorus*, showed that *C. crescentus* ghost cells were stained entirely, like uninfected cells, indicating that predation does not massively disrupt the prey cell wall (Fig. 4f). These observations also imply that prey cells stop growing upon contact with the predator. Higher resolution imaging by cryo-EM indicated the presence of a complete outer membrane (OM) and S-layer at different stages during predation, including on the *C. crescentus* post-feeding ghost cells (Fig. 4d, e). Thus, these data demonstrate that the three outer layers of the prey envelope (i.e., S-layer, OM, and PG) are not visibly altered by the predation process. However, the *C. crescentus* inner membrane (IM) progressively loses its integrity (Fig. 4d, magenta arrowheads and Fig. 2d), suggesting that feeding is accompanied by the injection of lipolytic activity in the prey. At early predation stages, when the granularity of the prey cytoplasm is still dense, localized IM ruptures were observed, while only dispersed, collapsed IM fragments were found in the prey cell during later stages of *B. exovorus* growth and upon detachment of the last predator cell from the *C. crescentus* ghost (Fig. 4d).

## Cryo-electron microscopy hints at a fixed-size junction securely joining prey and predator cells

The observed disruption of the prey IM might facilitate the uptake of digested cytosolic components into the *B. exovorus* cell, whereas keeping outer layers unharmed might prevent leakage of the prey content into the medium, and/or shield the feeding predator from the proteolytic, nucleolytic, and lipolytic activity in the prey. Thus, we expected the need for a secure and firm attachment site between the *B. exovorus* and the *C. crescentus* cells to avoid the spillage of the prey cytoplasmic content and assure a continuous fastening and nutrient flow throughout the *B. exovorus* cell cycle. This idea is supported by cryo-EM images of the contact between the *B. exovorus* and *C. crescentus*. The outer layers of the prey (OM and S-layer, when using the wild-type CB15N) appear to be pulled towards the predator OM, forming a "feeding" junction (Figs. 5a, b, 2d). We measured that this junction features a remarkably consistent diameter of 165 ± 20 nm (*n* = 48 cells, from two independent cryo-EM image acquisitions) (Fig. 5c), which is similar when *B. exovorus* is attached to Δ*rsaA C. crescentus* cells (164 ± 22 nm, *n* = 12 cells) (Fig. 5c and Supplementary Fig. 4). These observations suggest the assembly of a dedicated macromolecular clamp that might permit the selective import of nutrients into the predator cell. Importantly, such a complex must be disassembled in a way that detachment of *B. exovorus* from its prey does not impair the integrity of the predator cell envelope.

## Discussion

In this study, we used a combination of live-cell imaging and cryo-electron microscopy to reveal for the first time the complete lifecycle of an epibiotic predatory bacterium as well as the fate of its prey (Fig. 6). We first discovered that *B. exovorus* is capable of predating *C. crescentus* prey cells that carry an S-layer, visualized by cryo-EM as a continuous thin lattice outside the OM and surrounding the entire cell as previously characterized[22]. The presence of the S-layer does not significantly delay predator attachment or killing kinetics. Additionally, the size of the feeding junction between the predator and its prey is unaltered by the presence or absence of the S-layer. Although the physiological function of the S-layer is highly diverse across species, it was proposed to act as a protective shield against predatory bacteria based on experiments using *Bdellovibrio* predators and various species and strains as prey[18]. However, our observations that the S-layer does not provide protection against *B. exovorus* predation imply that other factor(s), yet to be determined, may play a role in protecting some strains or species from predation. Along with this idea, it has been shown that *B. exovorus* exhibits a limited prey range, primarily targeting the α-proteobacterium *C. crescentus*[10,18]. Only weak lysis was reported for a few other species in the presence of the *B. exovorus* type strain[11]. By subjecting a range of α-proteobacterial species to *B. exovorus* predation, we demonstrated that, within the α-proteobacterial class, its prey range extends beyond *C. crescentus* but is confined to specific families. The molecular reason for this predator–prey specificity and whether it represents the need for specific attachment sites on the prey, or rather a resistance mechanism in non-prey species, remains unclear. Phylogenetic analysis unveiled at least one evolutionary event where Rhizobiaceae and Brucellaceae might have lost sensitivity or developed a defensive mechanism against *B. exovorus* predation. Interestingly, none of the tested species, including those not predated by *B. exovorus*, are known to carry an S-layer. Further investigation into the family-specific envelope composition of α-proteobacteria might uncover specific characteristics that provide defense or sensitivity to epibiotic predation.

By using live-cell microscopy, our study also reveals that *B. exovorus* uses non-binary division to adopt a unique proliferation mode. We observed that *B. exovorus* mainly produces triplet progenies while firmly attached to the prey cell surface, which was unnoticed in previous transmission electron microscopy observations[10,11,28]. Our time-lapse images not only confirm the epibiotic lifestyle of *B. exovorus* but also provide insights into a novel division pattern, wherein a cell elongates and subsequently releases progenies sequentially from its distal end. While this behavior is reminiscent of budding, it is noteworthy that the predator cell that is directly attached to the prey does not initiate new rounds of growth and division once the first progenies have left, but instead, departs from the prey surface to resume the lifecycle from the attack phase. Furthermore, the quantification of offspring number showed that *B. exovorus* produces "fixed" numbers of progenies, typically three and less frequently two, irrespective of the prey size. This contrasts with the behavior of the closely related endobiotic predatory bacterium *B. bacteriovorus*, which adjusts its growth phase to prey variability,

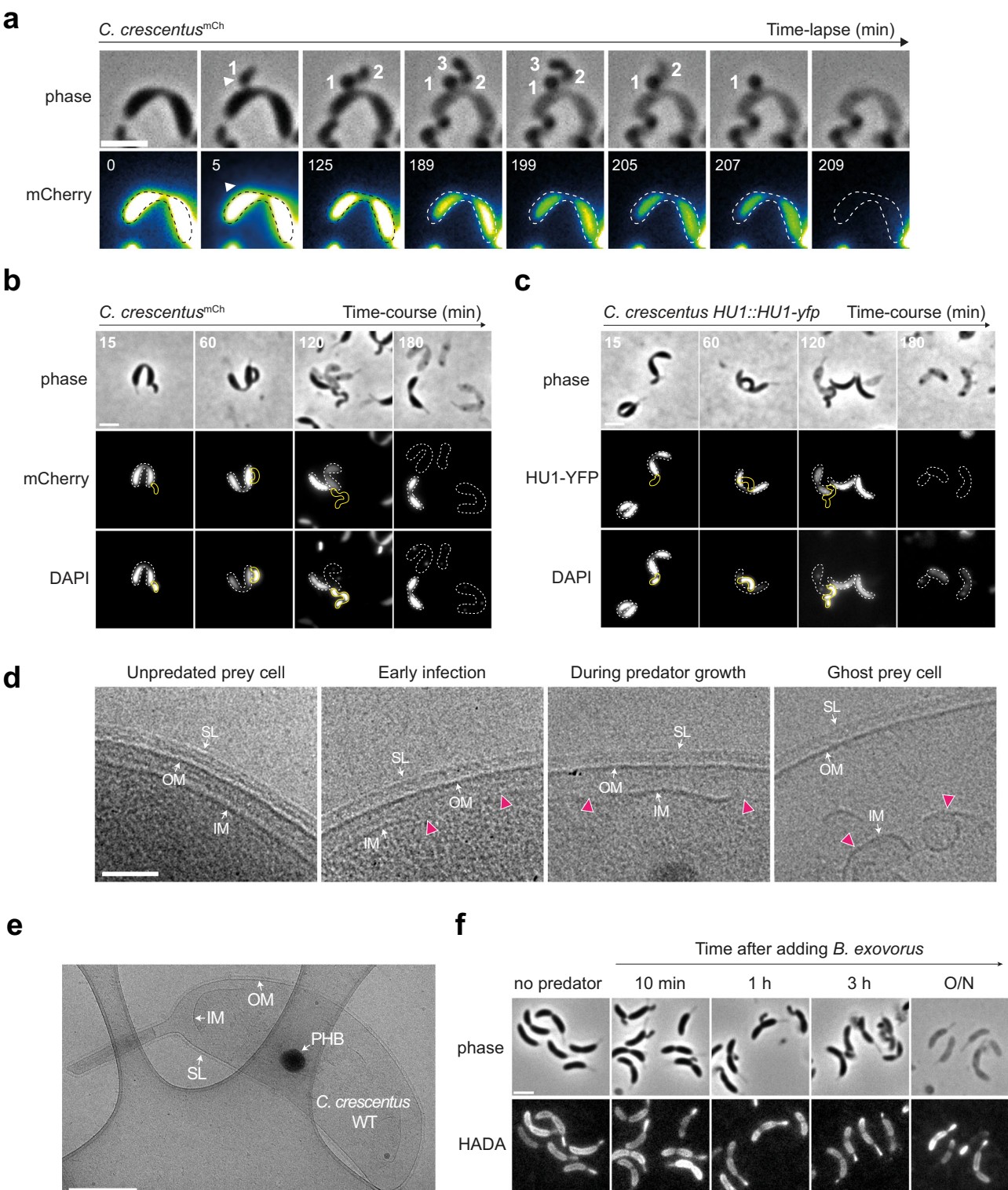

resulting in the scaling of progeny number with prey cell size[32]. It is conceivable that growing inside a prey provides a protective nest, enabling the exploitation of all available prey resources, if the predator can adjust its cell cycle to the variability of the prey—which is the case with *B. bacteriovorus*. In contrast, an external feeding mode could represent a challenge due to exposure to external perturbations such as other predators, bacteriophages, or changes in the physico-chemical properties of the environment. This might explain why *B. exovorus* produces a limited number of progenies without

adaptation to prey dimensions, despite relying on non-binary division for proliferation, which might represent a common trait among *Bdellovibrio* species.

Our data suggest that the type of predation could also impact the feeding profile of the prey. Unlike *B. bacteriovorus*, which does not seem to require prey IM disruption for feeding (a shrinking but clearly defined prey cytoplasm can be seen throughout the predator growth phase; e.g., refs. 31,32,37), predation by *B. exovorus* quickly results in the loss of prey IM integrity, possibly facilitating digestion and/or

**Fig. 4 | *B. exovorus* digests prey content in situ through unaltered external envelope layers. a** The mCherry fluorescent signal is used as a reporter of the proteinaceous cytoplasmic content. Representative time-lapse microscopy images of the mCherry-producing *C. crescentus* (*C. crescentus^mCh*) predated by *B. exovorus*. The number of future predator daughter cells is indicated on the phase contrast images. The fluorescence signal was false colored with the GreenFireBlue colormap in Fiji to display changes in fluorescence intensity. The white arrowhead points at the *B. exovorus* attached to the prey surface. The predated *C. crescentus* cell outlines shown as dashed lines were drawn manually based on the phase contrast image at time 0. Scale bar, 2 µm. **b, c** Time-course imaging of the *C. crescentus^mCh* (**b**) or the *C. crescentus HU1::HU1-yfp* (**c**) strain predated by *B. exovorus*. Predator cells were mixed with preys and stained with DAPI 10 min prior imaging at each selected time point. Top: phase contrast, middle: mCherry (**b**) or HU1-YFP (**c**) signal, and bottom: DAPI fluorescence images of selected time points from a representative experiment are shown. Cell outlines for *B. exovorus* (yellow) and *C. crescentus* (dashed white)

were drawn manually based on phase contrast images. Scale bar, 2 µm. **d** Representatives cryo-EM images of the *C. crescentus* CB15N envelope layers and cytoplasm during *B. exovorus* predation. Partial or total disruptions (magenta arrowheads) of the prey's inner membrane are observed through the predatory cell cycle. SL S-layer, OM outer membrane, IM inner membrane. Scale bar, 0.1 µm. **e** Representative cryo-EM image of an entire *C. crescentus* ghost cell. The integrity of the outer layers (S-layer and outer membrane) is conserved, maintaining the original cell shape. PHB polyhydroxybutyrate granule, SL S-layer, OM outer membrane, IM inner membrane. Scale bar, 0.5 µm. **d, e** Experiments were repeated twice with similar results. **f** Phase contrast (top) and fluorescence (bottom) images of wild-type *C. crescentus* labeled with the fluorescent D-amino acid HADA for 3 h, at the indicated time points before or upon addition of *B. exovorus*. Brightness and contrast are adjusted individually for each time point for display purposes. Predated and ghost *C. crescentus* cells retain HADA labeling all around the cells. Scale bar, 2 µm.

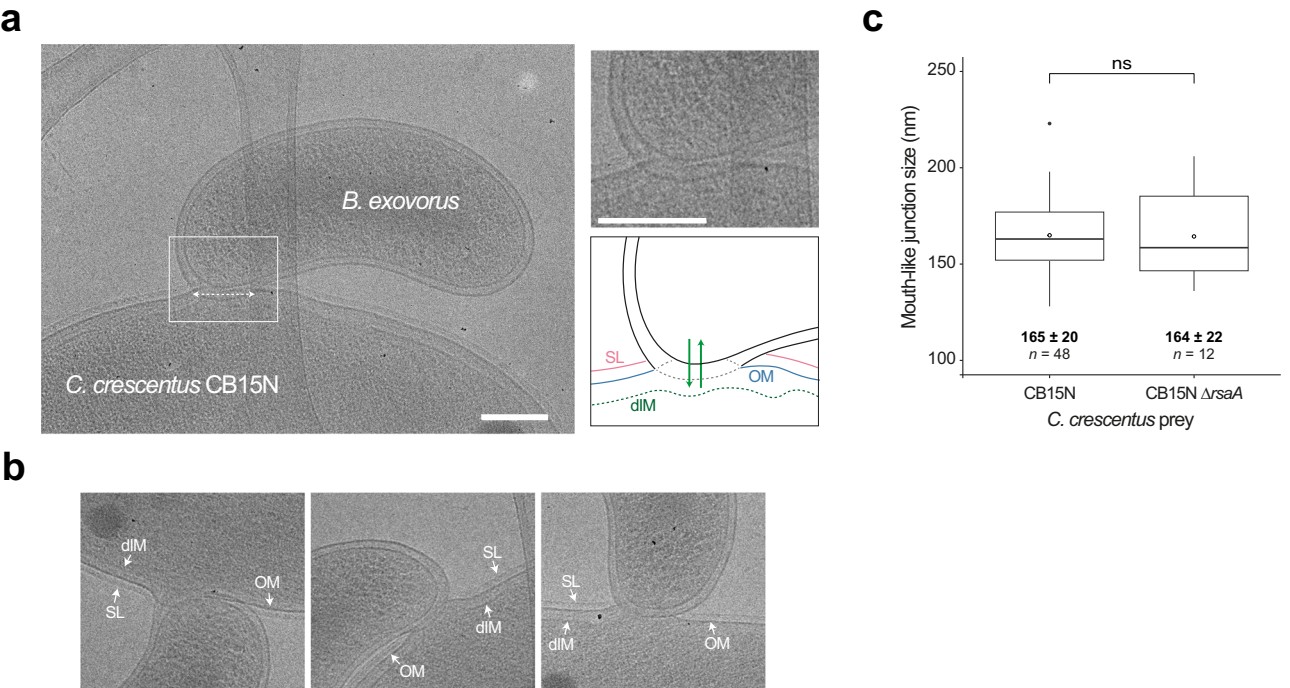

**Fig. 5 | Predator feeding occurs through a fixed-size junction clamping prey and predator outer membranes. a** Representative cryo-EM image of *B. exovorus* attached to the wild-type *C. crescentus* cell surface. The dashed double-arrow highlights the fixed-size junction between the predator and prey outer membranes. This image was previously used in Fig. 4d to show the early prey inner membrane disruption (Early infection). No S-layer structure is visible on the predator surface. A magnified view of the contact site is displayed in the top-right corner, accompanied by a hand-drawn schematic representation based on the cryo-EM image. Green arrows represent the export of *B. exovorus* molecules in the prey and the import of prey content during feeding. SL S-layer (pink), OM outer membrane (blue), dIM

disrupted inner membrane (green). Other examples are shown in **b**. Scale bar, 0.2 µm. **c** Boxplot representation of the size of the predator–prey junction with the wild-type or the Δ*rsaA C. crescentus* strain as a prey. Bold horizontal bars represent the median value; empty circles represent the mean; the lower and upper boundaries of the internal box plot correspond to the 25th and 75th percentiles, respectively; the whiskers represent the 10th and 90th percentiles. Values of the mean, the standard deviation, and the number of analyzed cells (*n*) are indicated. A pairwise comparison from two biological replicates is indicated above the plot (ns nonsignificant; two-sample Fisher–Pitman permutation test). Source data are provided as a Source Data file.

import of nutrients from the cytoplasm. Our results also suggest that the prey cellular content is digested in situ by *B. exovorus*. While the import pathways and the preferential nutrients promoting predator growth remain to be determined, our observations imply that early-secreted *B. exovorus* effectors degrade at least the DNA and protein content. This is consistent with the lack of complete biosynthesis pathways for several essential amino acids, and the relative abundance of putative peptidases encoded in the *B. exovorus* genome[28]. The sudden disappearance of remaining cytosolic proteins when the last predator cell leaves the prey, together with the cryo-EM observations of ghost cells devoid of cytoplasmic content, indicates that *B. exovorus* does not reseal the prey upon detachment.

Ultimately, we showed that *B. exovorus* establishes a feeding junction that tightly connects predator and prey outer membranes during the whole predator growth phase. This is at odds with the epibiont *Vampirococcus lugosii*, which proliferates while consuming the content of the host cell from the surface, but whose membrane remains separated from the host cell membrane by a relatively large space[7]. Whereas this junction is specifically located at the non-flagellated pole of *B. exovorus*, which we propose to designate as "feeding pole", there was no preferential landing zone for predators on the prey cell surface. We think that a mere fusion between the outer membranes of two different cells is highly unlikely, as it might lead to the uncontrolled and potentially deleterious integration of

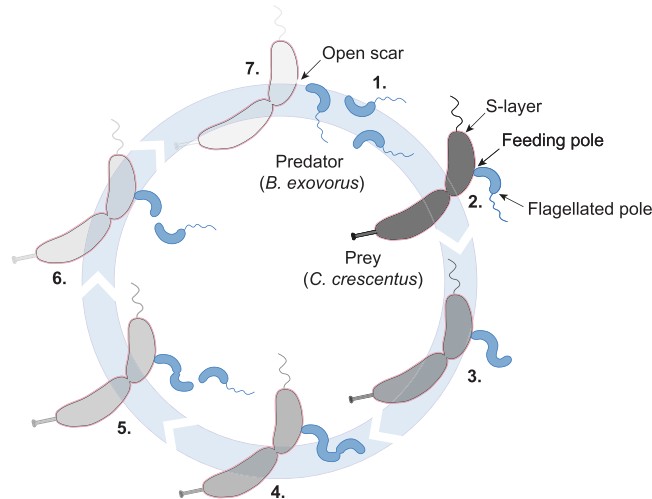

**Fig. 6 | Model of the *B. exovorus* life cycle.** Numbers indicate key steps in the cycle: (1) *B. exovorus* predator initially hunts for its prey. (2) Once tightly attached to the prey cell surface, the predator forms a specific junction at the feeding pole (opposite to the flagellated pole), pulling the prey's outer layers (S-layer in pink and outer membrane in dark gray) towards its outer membrane. (3) The secretion of specific enzymes enables the in situ digestion of prey contents. Degraded prey molecules are then imported, promoting predator elongation. (4) As the predator grows, constriction sites sequentially emerge along the filament. At the end of the filament growth, cell division sequentially releases the outermost (5) and then the second progeny within minutes (6). (7) Finally, the last predator cell detaches from the prey surface, leaving a ghost cell devoid of its cellular contents but maintaining its shape. The escape of the mother cell leaves an open scar that is not resealed by *B. exovorus*.

components from the prey OM into the predator envelope. This idea is supported by our cryo-EM analysis which did not indicate the transfer of the prey outermost layer (S-layer) to the predator at any predation stage. Moreover, the remarkably constant and relatively large diameter of this feeding junction (~160 nm, compared to an average cell length of ~1.2 μm) strongly suggests that this junction could represent a specialized platform for the prey-predator interaction. This platform likely accommodates different types of transporters that regulate trafficking, allowing two distinct but complementary activities: the secretion of digestive enzymes and the import of nutrients, promoting predator growth. Notably, our data suggest that the junction is resorbed unilaterally at the end of the predatory cycle, leaving an open scar in the prey but an unaltered feeding pole on the predator side, allowing it to resume hunting. The molecular nature of the complexes potentially hosted in this platform and the mechanisms by which they could be temporally and spatially regulated in coordination with the predatory lifestyle represent intriguing avenues of investigation. The development of genetic tools to manipulate *B. exovorus* and comparative genomics with the endobiotic predator *B. bacteriovorus* will be crucial to investigate the molecular determinants behind the different division cycles of these closely related predators, and which mechanisms allow them to feed either from inside or outside the prey.

## Methods

### Bacterial strains and growth conditions
All strains used in this study are listed in Table S1. *B. exovorus* JSS (taxon: 453816; ATCC-BAA-2330)[10] was used as a model predator strain. *Caulobacter crescentus* CB15N[38] and its derivatives were grown in PYE (rich medium) at 30 °C with aeration. *Ochrobacterum anthropii*, *Agrobacterium tumefaciens*, and *Sinorhizobium meliloti* were grown in an LB (rich medium) at 30 °C with constant shaking. *Asticcacaulis excentricus*, *Asticcacaulis biprosthecum*, *Asticcacaulis benevestitus*, and

*Brevundimonas subvibrioides* were grown in PYE at 30 °C and constant shaking. *A. benevestitus* growth medium was supplemented with 0.05% xylose. *Phenylobacterium lituiforme* was grown in PYE at 37 °C and *Hyphomonas neptunium* was grown in Marine broth at 30 °C and constant shaking. The *B. exovorus* strain was routinely grown in M2 buffer (17.4 g.L$^{-1}$ Na$_2$HPO$_4$, 10.6 g.L$^{-1}$ KH$_2$PO$_4$, and 5 g.L$^{-1}$ NH$_4$Cl) supplemented with 0.5 mM MgSO$_4$, 0.5 mM CaCl$_2$, and 0.1X ferrous sulfate chelate solution (Sigma-Aldrich) (M2$^+$) with stationary-phase *C. crescentus* CB15N or the related Δ*rsaA* mutant as prey at 30 °C, and constant shaking. Note that predation was also efficient when using exponentially growing *C. crescentus* CB15N or CB15N Δ*rsaA*. *B. exovorus* attack phase cells were then isolated from the prey lysate by filtration through a 1.2-μm syringe filter. We acknowledge the current technical limitation of not being able to precisely quantify the engaged predator cell numbers prior to the experiments. Therefore, predator inoculum volumes were determined empirically (see below). Moreover, experiments comparing predation on different prey strains were always conducted in parallel, using the same predator starter suspension and inoculum volume.

### Bacterial strains and plasmids construction
The *C. crescentus*$^{mCh}$ strain was obtained by conjugation between CB15N and *E. coli* S17-1 λpir carrying the pXbiofab-mcherry mobilizable plasmid, which allows the constitutive expression of *mcherry* from the synthetic promoter Pbiofab and includes a 2-kb homology region for recombination and one-step integration of the vector at the *xylX* locus. Transconjugants were selected on kanamycin. To construct pXbiofab-mcherry, the inducible P$_{xyl}$ promoter was removed from the pXGFPC-2 vector[39] by PCR with primers oGL2113 (5′-gctagctgcagcccgg-3′) and oGL2117 (5′-cgacaaaccacgacctggacc-3′). The linearized vector was assembled using the HiFi DNA assembly mix (New England Biolabs) with a Pbiofab-mcherry fragment, amplified by PCR from pSEVA251-pbiofab-mcherry using primers oGL2118 (5′-aagaggtccaggtcgtggtttgtcggcatcgatagagtattgacttcgcatctttttg-3′) and oGL2116 (5′-gtggatccccccgggctgcagctagcattctcaccaataaaaaacgcccggcg-3′). To construct pSEVA251-pbiofab-mcherry, the HiFi DNA assembly mix (New England Biolabs) was used to assemble the mcherry fragment, amplified by PCR from pVCHYC-6 (from ref. 39) using primers oGL887 (5′-AATTCAGGGTGGTGAATATGGTGAGCAAGGGCGAG-3′) and oGL925 (5′-catgcctgcaggtcgacttaCTTGTACAGCTCGTCCATGC-3′), and the pBG18 vector (pSEVA251-pbiofab-sfGFP; constructed by Sarah Bigot, kind gift from Christian Lesterlin), linearized by PCR using primers oGL783 (5′-CATATTCACCACCCTGAATTGACTCTCTT-3′) and oGL784 (5′-gtcgacctgcaggcatgc-3′) to remove the sfGFP fragment.

### Phylogenetic tree construction
The phylogeny of selected species was derived from 16S rRNA sequences. The sequences were obtained from the Ribosomal Database Project or the NCBI Database and aligned using MUSCLE[40]. The maximum likelihood tree was then generated using RaxmlGUI 2.0 and the resulting tree was formatted using iTOL.

### Prey killing kinetics assay
To assess predation efficiency, equal amounts of predators from the same fresh filtered lysate were mixed with the wild-type or the Δ*rsaA C. crescentus* CB15N at a final OD$_{600}$ of 1, and M2 buffer was added to reach 300 μL per well in a transparent 96-well flat bottom plate. Technical triplicates were prepared in separate wells of the same plate in each experiment. The plate was shaken continuously (frequency 567 cpm (3 mm)) at 30 °C for 16 h in a Synergy H1m microplate reader (Biotek). Optical density measurements at 660 nm were taken every 20 min. A decrease of OD$_{660}$ indicates prey lysis, as *B. exovorus* cells do not affect absorbance. Predation kinetics metrics were extracted using CuRveR as previously described[41].

## Phase contrast and fluorescence microscopy

Phase contrast and fluorescence microscopy images were acquired on a Ti2-E fully motorized inverted epifluorescence microscope (Nikon) equipped with a CFI Plan Apochromat λ DM 100×1.45/0.13 mm Ph3 oil objective (Nikon), a Sola SEII FISH Illuminator (Lumencor), a Prime BSI camera (Photometrics), a temperature-controlled and light-protected enclosure (Okolab), and filter cubes for DAPI (32 mm, excitation 377/50, dichroic 409, emission 447/60; Nikon), mCherry (32 mm, excitation 562/40, dichroic 593, emission 640/75; Nikon), and GFP (32 mm, excitation 466/40, dichroic 495, emission 525/50; Nikon). Image acquisition was controlled by the NIS-Ar software (Nikon).

## Live imaging by phase contrast and fluorescence microscopy

Before imaging experiments, *B. exovorus* predator cells were grown overnight on *C. crescentus* CB15N (or CB15N Δ*rsaA* when indicated) and passed through a 1.2 μm filter to eliminate most remaining (uninfected and ghost) preys. *C. crescentus* prey cells were grown in PYE until the late stationary phase (OD$_{600}$ = 1.2–1.6), harvested at 2600×*g* at room temperature (RT) for 4 min, washed twice, and resuspended in M2$^+$ medium. For time-lapse imaging of predation, this suspension of *C. crescentus* prey cells was diluted (OD$_{600}$ = 0.02) and flowed into an Ibidi® microfluidics chamber sealed with a biocompatible chitosan-coated glass coverslip (Chitozen, Idylle)[42] to promote cell adhesion, following manufacturer's instructions. The microfluidics chamber was then subjected to four washes with M2$^+$ to remove unattached cells before being positioned under the microscope. Time-lapse imaging started prior to flowing the microfluidics chamber with a filtered culture of *B. exovorus*. The predator-to-prey ratio was empirically adjusted to about 1:1 prey:predator, which both allowed synchronized predation (i.e., most prey cells are infected by a predator at the same time) and subsequent single-cell analyses. We consider the prey-predator mixing step as the time 0 in all our predation imaging experiments. Since most prey cells were targeted by at least one *B. exovorus* cell within the first few minutes, we consider the imaged predatory growth as synchronous. The same fields of view were imaged at 5-minute intervals (for the first 2 h) and then at 2-min intervals. The enclosure temperature was set to 28 °C.

For snapshots of predator–preys mixes, cells were spotted on 1.2% agarose pads prepared with M2$^+$ medium after overnight co-incubation or at different time points upon mixing prey and predator (as above), as mentioned in figures. When indicated, the fluorescent dsDNA-labeling dye 4′,6-diamidino-2-phenylindole (DAPI, Thermo Fisher) was added to the culture at a concentration of 5 μg.mL$^{-1}$ for 10 min prior to imaging. For labeling of the *C. crescentus* peptidoglycan, a starter culture of the wild-type *C. crescentus* CB15N was grown overnight in PYE at 30 °C, and diluted in PYE to reach early exponential phase after overnight incubation at 30 °C. When OD$_{600}$ reached ~0.1, the blue fluorescent D-amino acid HADA (Bio-Techne, reference #6647/5) was added to a concentration of 250 μM (from a 100 mM stock solution in DMSO). Cells were incubated at 30 °C with shaking and protected from light for 3 h (allowing the labeling of the entire cell wall) before direct loading and immobilisation into the chitosan-coated chamber as described above. The chamber was washed four times with M2$^+$ in the dark, immediately before the start of image acquisition.

## Sample preparation and cryo-EM images acquisition

For cryo-EM data acquisition, sample grids were prepared by spotting 3 μL of the predation mix (*B. exovorus* with *C. crescentus* CB15N wild-type and Δ*rsaA*, harvested at different times upon mixing) on freshly glow-discharged Lacey grids (300 Mesh Cu−Agar scientific), manually back blotted for 3−4 s and flash-frozen in liquid ethane using a CP-3 plunger (Gatan). Data were collected on a 300-kV CRYO ARM™ 300 (JEM-Z300FSC) Field Emission Cryo-Electron Microscope (JEOL) equipped with a K3 Summit direct electron detector (Gatan) at the

VIB-VUB Bio-Electron Cryo-Microscopy center (BECM) in Brussels, Belgium. We collected 2 by 2 montages using image shift at a magnification of 5 k (10.2 Å per pixel) at a dose of 22.7 e- per pixel per second in counting mode, with a total exposure of 4.99 s. Images were collected with a defocus of −5 to −10 μm.

## Image analysis

For the measurement of cell morphology parameters of attack phase *B. exovorus* cells, cell outlines were obtained with Oufti[43] from phase contrast images of filtered overnight mix between *B. exovorus* and the indicated *C. crescentus* strain. Cell length and cell area were extracted and plotted in Matlab using the home-made histCellArea.m and histCellLength.m Matlab scripts from[44] available on the lab GitHub page (https://github.com/geraldinelaloux/Kaljevic-et-al-2023). The demograph of DAPI-stained cells was obtained using the built-in demograph function in Oufti[43]. The size of feeding junctions was defined as the distance between the opposite predator outer membrane attachment sites (as indicated on Fig. 5a) and was estimated using the measuring tool in ImageJ (version 1.53t). Figures were prepared with Adobe Illustrator (Adobe).

## Image processing

Images were processed with Fiji[45] to prepare figures, using identical settings for fluorescence contrast and brightness across all regions of interest within the same figure panel unless indicated otherwise. Figures were assembled using Adobe Illustrator (Adobe, Inc.).

## Statistics and reproducibility

All analyses of microscopy images were performed using several representative fields of view from at least three independent biological replicates. Means and standard deviations were calculated in R using the base R functions. The normality of the data was assessed using the Shapiro test. Nonparametric pairwise comparisons of datasets were performed using the unpaired Wilcoxon rank-sum test or the two-sided two-sample Fisher–Pitman permutation test. Significance was defined by $p > 0.05$ (ns), $p \leq 0.05$ (*), $p \leq 0.01$ (**), and $p \leq 0.001$ (***).

## Reporting summary

Further information on research design is available in the Nature Portfolio Reporting Summary linked to this article.

## Data availability

The data that support the findings of this study are available from the corresponding author on request. Source data are provided with this paper.

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

## Acknowledgements

We thank the VIB-VUB facility for Biological Electron Cryogenic Microscopy (BECM) and Marcus Fislage for technical assistance. We are grateful to Régis Hallez for providing the CB15N Δ*rsaA* strain, Yves Brun for the kind gift of *Asticcacaulis*, *Brevundimonas*, *Hypohomonas*, and *Phenylobacterium* strains, Xavier De Bolle for sharing *Agrobacterium*, *Ochrobactrum*, and *Sinorhizobium* strains, and Christine Jacobs-Wagner for providing the HU1-YFP *C. crescentus* strain. We thank Antonella Fioravanti for an insightful discussion on S-layers, Sander Govers and Joel Hallgren for advice regarding bacterial phylogeny, the Laloux lab and Michaël Deghelt for fruitful discussions and thorough reading of the first draft, and Charles de Pierpont for excellent technical assistance. Y.G.S. was a post-doctoral fellow of the European Molecular Biology Organization (EMBO) and is

currently a post-doctoral fellow of the F.R.S.-FNRS, A.S. is a post-doctoral fellow of the European Molecular Biology Organization (EMBO) and the Marie Skłodowska-Curie Actions (MSCA), Y.B. is a recipient of a post-doctoral scholarship from the Wallonia-Brussels International Excellence Grants Program (IN WBI), G.L. is a Research Associate of the F.R.S.-FNRS. This work received support from an Actions de Recherche Concertées (ARC) grant from UCLouvain (PAChIDERM, to G.L.). Work in the Laloux lab is supported by the European Research Council (ERC) under the European Union's Horizon 2020 research and innovation program (ERC Starting Grant PREDATOR #802331).

## Author contributions

Y.G.S. and G.L. performed live imaging experiments, Y.G.S. performed statistical analyses, image analysis, and prepared all the figures; Y.B. contributed to the optimization of *B. exovorus* experiments; A.S. prepared the cryo-EM grids and performed quantification from cryo-EM images; A.S. and H.R. performed the cryo-EM image acquisition; Y.G.S., A.S., H.R., and G.L. analysed the cryo-EM images; Y.G.S. and G.L. wrote the first draft; all authors revised the manuscript; G.L. supervised the work.

## Competing interests

The authors declare no competing interests.
