## [Peer Review File · Nature Communications]

Lifecycle of a predatory bacterium vampirizing its prey
through the cell envelope and S-layerReviewer #1 (Remarks to the Author):

Santin et al., investigate how the epibiotic *B. exovorus* predate *C. crescentus*, a known prey for this organism. Using time-lapse microscopy, they find that *B. exovorus* can predate *C. crescentus* independent of its S-layer. They further show that the number of progenies produced (on average 3) does not scale with cell size, and that there appears to be no preference for where *B. exovorus* may attach to the *Caulobacter* cell. They apply cryo-EM microscopy to image the attachment junction of the predator on the prey surface and conclude that the OM and S-layer are intact during the predation process and it is likely that *B. exovorus* digests the content of the prey and detaches, leaving behind an open scar in *Caulobacter*. *B. exovorus* displays some prey specificity and the mechanism of attachment, as well as the specificity remain open questions.

This is exciting and novel work on exploration of bacterial predator-prey cell biology. Although phenomenological, the findings described here provide a solid and important frame work to further study the molecular mechanisms underlying the observations on the interactions between *B. exovorus* and *C. crescentus*. The work is well-done and the writing is also clear. I have only a few comments below that the authors should address prior to publication:

- a. The attachment frequency (Fig. 2C) is calculated at a timepoint that falls at the beginning of the curve shown in Fig. 2B. At this timepoint, no difference is seen between the two strains. However, there appears to be some difference between the *rsaA* deletion and wt strain over time in Fig. 2B (for example, after 4 h). Can the authors conduct a time course for the measurement of the attachment frequency, sampling times along the curve in Fig. 2B? Ideally, a competition assay between the two strains (treated with predator) would be a clear experiment to support the conclusion that the S-layer does not play a role in protection.
- b. The authors use Fig. S2d to conclude that the growth dynamics of *B. exovorus* remains the same between *rsaA* deletion and wt. However, in the example provided, two *B. exovorus* appear to have attached to one *Caulobacter* while in the wt example the ratio appears to be 1:1. Is this correct? Authors should provide a quantification of the prey per predator cell in wt and *rsaA* deletion? This can also be done for the *popZ* deletion.
- c. L200 and associated figure. Can the authors rule out protein unfolding due to environmental changes such as pH change, as opposed to digestion of prey protein? To make this conclusion, some other assay will also have to be used (such as mass spec measurement of total protein content of prey population during predation. The section could be rewritten to include other possibilities for the observation of mCherry signal loss.
- d. Discussion: could the authors perhaps speculate on what constitutes the attachment junction? Are there hints of some protein complexes that play a role in the same, based on the genes encoded by *B. exovorus*?

Minor comments:

- a. Figure legends: The number of experimental repeats and total number of cells imaged should be provided in all legends.
- b. L57: against protozoa, nematodes and bacteriophages
- c. L109: could the buffer conditions be included in the main text, or the sentence rewritten for clarity
- d. In the example provided to suggest that *B. exovorus* can attach to the stalk as well (Fig. S1B), it appears that the predator is attached to both the stalk and the cell body. To clearly support 'stalk-only' attachment, another example should be included.
- e. L132: how was attachment frequency calculated?
- f. L158: Fig. 3a is referred to support formation of only two progenies. However, the highlighted cell appears to have three progenies that become two at T286min. Perhaps another example should be provided.
- g. L247: such 'a' complex
- h. L454: predation 'of'
- i. Fig. S1 legend: depicted 'in' Fig. 1a
- j. Fig. S2 legend: d-e reference for *rsaA* deletion and *popZ* deletion needs to be corrected

Reviewer #2 (Remarks to the Author):

The manuscript describes the lifecycle of *B. exovorus*, revealing predation through an S layer and a novel cell division system. The quality of the microscopy is superb and the data presented supports the conclusions. I have only a few minor points:

Figure 1b The curves of predation appear to be different for the different prey strains with the mutant OD beginning to reduce before the wild-type, resulting in a different S value although the r_{max} rates look similar after this delay. Is this delay due to the presence of the S-layer? The S and r_{max} values for each biological repeat should be presented and statistical analysis applied to these values. There may well be a statistically significant difference in S that should be commented upon.

Lines 185-187 There are statistically more occurrences of 4 progeny when preying upon the larger cells and whilst rare, this does suggest that progeny numbers are affected by larger size, albeit not to the extent shown for *B. bacteriovorus*.

Given that the relationship between numbers of progeny, progeny size and prey size don't equate in the same way as for *B. bacteriovorus*, was there any other differences when preying upon larger prey, for example numbers of storage granules?

Line 323 should this read "a different type" or "different types"?

Line 372 How many predators were added? Pfu?

Lines 431-36 How was normality tested for?

Line 494 representative

Reviewer #3 (Remarks to the Author):

The work presented in the manuscript by Santin et al is of interest to a wide audience. The identification of key components of the cell cycle of *B. exovorus* during predation is exciting and original work. Bacteria predator-prey interactions remain poorly understood and this work provides insights about the diversity of strategies/lifecycles of the predator bacteria. Most of the conclusions have data to support them with well thought-out methodologies and experimental plans. I only have a few comments. Overall, I enjoyed reading this manuscript.

Introduction section: More background information about *B. exovorus* is necessary. For instance, does B.e. have an S-layer? Figure 5 suggest that B.e. has no S-layer. Is that correct? Is B.e. an obligate epibiotic or can they grow in the absence of a bacterium prey?

Line 161 – 164. I am not convinced the data supports the conclusion that "B. exovorus initiates cell constriction during growth (Fig. 3a)..." The data on Figure 3a are only a couple sets of cells that seem to constrict but the figures are too small to really deduce anything. This potential constriction during growth should be quantified. The delta-popZ filamentous cells could facilitate the quantification of this observation. On Figure 3C, the sequential division in the pre-divisional cells in the cryo-EM images is clear. However, it is possible based on the size of those cells that they were done growing and only then initiated division sequentially.

The data collected from time lapses could be used to deduce a couple other important parameters. For instance, does the prey size have any correlation with the number of B.e. that adhere? Can multiple B.e. prey on a single *Caulobacter* bacterium? Is there a fixed time of predation? In bigger cells (delta-popZ), was the predation time longer than in smaller wildtype *Caulobacter* cells? A constant time of preying would support the comments given on lines 283 – 286.

The results section or at least the discussion section would benefit from a more thorough analyses of the bacterial species that are presented on Figure 1BC. Based on the outer membrane/cell

envelope genes found between susceptible and resistant species, is there anything that could explain the differences in susceptibility to B.e.? Based on bioinformatic analyses, is there anything that the authors could predict would cause the susceptibility? This information could broaden the impact of the findings and provide a big picture to the discussion section.

Reviewer #4 (Remarks to the Author):

Santin et al. investigate the life cycle of the predatory bacterium *Bdellovibrio exovorus* and the fate of its host bacterium *Caulobacter crescentus* during the infection cycle using optical microscopy coupled with cryo electron-microscopy (cryo-EM). The authors show that *B. exovorus* can grow on a restricted range of alpha-proteobacteria and even infect a *C. crescentus* wild-type strain expressing an S-layer, challenging a previous statement suggesting that the S-layer protects against predatory bacteria. Life-cell imaging data demonstrate that *B. exovorus* preferably reproduce triplet progenies independent of the host cell size. Overall, the experiments are well designed, the data is professionally documented, and the manuscript is well-written. I have a few suggestions which should further strengthen the manuscript.

The authors use a cytosolic mCherry *C. crescentus* strain to investigate the fate of the host cytoplasm. Loss of fluorescent intensity signal is interpreted as digestion/degradation of the host cytoplasm through *B. exovorus*. The infected host cells are imaged for 2 hours at every 5 intervals and then in 2 minutes intervals (Methods, lines 394-395). Photobleaching of cytoplasmic mCherry might partially contribute to the loss of fluorescent intensity signal over time. Therefore, I think it would be useful to image uninfected control host *C. crescentus* cells under identically imaging conditions and compare the fluorescent signal over time to the existing data of infected *C. crescentus* cells.

Lines 299-309: The authors suggests that the host cytoplasm is actively been degraded by secreted protease, nucleases, lipases from *B. exovorus*. Could the author comment or discuss if the *B. exovorus* genome encodes these effectors?

Lines 311-320: The authors suggests that the outer membrane of *B. exovorus* and its host *C. crescentus* do not fuse during the infection cycle based on lack of transfer of the S-layer from the host to the epibiont. The S-layer is known to bind to the O-antigen of the host lipopolysaccharide and neighbouring S-layer subunits. The lack of S-layer transfer to the epibiont might be due to strong interaction of RsaA subunits within S-layer lattice which prevents diffusion. Given the fast attachment of *B. exovorus* to host cells within 15 minutes (line 133), should it not be possible to pulse label host *C. crescentus* cells with a membrane dye such as FM4-64 (before internalisation of the dye)? The hypothesis can then be tested by time lapse-microscopy after mixing pulse-labelled host cells with the epibiont and investigating if the fluorescent signal remains within the host cell.

Regarding the point raised above, I think it would be useful to obtain a cryo electron tomogram of the "feeding pole" to show the discontinuity between the outer membranes of the epibiont and the host cell.

Minor comments:

Lines 112,343; Figure 1b,c: subvibriodes -> subvibrioides missing "i". (<https://www.ncbi.nlm.nih.gov/datasets/taxonomy/74313/>)

Line 133: "WT" -> wild-type (see line 135).

Line 164: Cryo-EM -> The abbreviation has not yet been specified.

Line 243: (n = 48 cells) -> n should be italicised.

Line 367: MUSCLE -> Reference/Citation missing.

Line 425: GitHub -> Please provide the link in the method section or data/software accessibility.

Lines 454/458: "α"-proteobacterial. The main text refers to "alpha"-proteobacteria (see line 88, 111, 120, ...). Use either α or alpha throughout the manuscript.

Line 481: (left) Scale bar, 0.5 μm -> Full stop missing between "(left)" and "Scale bar".

Line 509: Fiji -> The method section only refers to ImageJ (version 1.53t). Reference/Citation missing for Fiji or ImageJ.

Line 590: Cryo-EM imaging of predator:prey junction using the ΔrsaA *C. crescentus* -> *C. crescentus* should be italicised.

Response to the reviewers – Santin, Sogues, et al.

We thank all reviewers for their insightful comments and their positive appreciation of our work. Please find our point-by-point response and details of our revision below.

Reviewer #1

Santin et al., investigate how the epibiotic *B. exovorus* predaes *C. crescentus*, a known prey for this organism. Using time-lapse microscopy, they find that *B. exovorus* can predate *C. crescentus* independent of its S-layer. They further show that the number of progenies produced (on average 3) does not scale with cell size, and that there appears to be no preference for where *B. exovorus* may attach to the *Caulobacter* cell. They apply cryo-EM microscopy to image the attachment junction of the predator on the prey surface and conclude that the OM and S-layer are intact during the predation process and it is likely that *B. exovorus* digests the content of the prey and detaches, leaving behind an open scar in *Caulobacter*. *B. exovorus* displays some prey specificity and the mechanism of attachment, as well as the specificity remain open questions.

This is exciting and novel work on exploration of bacterial predator-prey cell biology. Although phenomenological, the findings described here provide a solid and important frame work to further study the molecular mechanisms underlying the observations on the interactions between *B. exovorus* and *C. crescentus*. The work is well-done and the writing is also clear. I have only a few comments below that the authors should address prior to publication:

a. The attachment frequency (Fig. 2C) is calculated at a timepoint that falls at the beginning of the curve shown in Fig. 2B. At this timepoint, no difference is seen between the two strains. However, there appears to be some difference between the *rsaA* deletion and wt strain over time in Fig. 2B (for example, after 4 h). Can the authors conduct a time course for the measurement of the attachment frequency, sampling times along the curve in Fig. 2B? Ideally, a competition assay between the two strains (treated with predator) would be a clear experiment to support the conclusion that the S-layer does not play a role in protection.

Response 1. We agree with the reviewer that there is a tendency to reach r_{max} slightly earlier in killing curves (Fig. 2b) when using the *rsaA* mutant as a prey (revised Supplementary Fig. 1b). Whereas we found no statistically significant difference in r_{max} and s values (see revised Supplementary Fig. 1a and 1b), this consistent trend in all replicates might suggest that the S-layer delays some steps of predation, such as prey detection, establishment of the junction, and/or the onset of growth. However, we think that this effect, which is observed only at the population level, is likely marginal, given that (i) the S-layer does not prevent predation, (ii) the killing speed given by r_{max} is comparable, (iii) attachment frequencies are similar between WT and $\Delta rsaA$ strains (Fig. 2c), (iv) *B. exovorus* forms a similar junction with both prey strains as see in cryo-EM (Fig. 5 & Supplementary Fig. 4), and (v) predation times are also similar (revised Supplementary Fig. 2f, see Response 23 to Reviewer 3). See also Response 15 to Reviewer 2.

Furthermore, as attachment is an early phase of the cycle, conducting a time-dependent attachment measurement would not be relevant, especially since our experimental setup favors synchronized predation cycles. We believe that a competition experiment between the two prey strains could address the question of whether a prey is preferentially infected by *B. exovorus* when both are present. However, such an experiment would not provide stronger support for the assertion that the S-layer does not protect from predation, given the explanations provided above. From a physiological standpoint, *B. exovorus* is unlikely to face the “choice” between prey with or without an S-layer in nature.

b. The authors use Fig. S2d to conclude that the growth dynamics of *B. exovorus* remains the same between *rsaA* deletion and et. However, in the example provided, two *B. exovorus* appear to have attached to one *Caulobacter* while in the wt example the ratio appears to be 1:1. Is this correct? Authors should

provide a quantification of the prey per predator cell in wt and *rsaA* deletion? This can also be done for the *popZ* deletion.

Response 2. We had selected the region of interest in Supplementary Fig. 2d (Supplementary Fig. 2e in the revised manuscript) to illustrate that *B. exovorus* can produce 2 or 3 progenies when using the *rsaA* mutant as prey, similar to predation on the wild-type prey. This image indeed includes a case of co-infection. However, such cases were similarly observed on the wild-type prey.

Co-infection seems primarily influenced by the amount of predator cells, mirroring our previous observations with *B. bacteriovorus* (Santin et al., *Curr Biol* 2023), but we acknowledge the limitation of not being able to precisely quantify the predator load in our current experimental setup. Consequently, in our live imaging experiments the predator-to-prey ratio is empirically adjusted to about 1:1 prey:predator, which also allows synchronized predation (i.e., most prey cells are infected by a predator at the same time), and enable single-cell analyses. We think that optimizing a method to precisely measure predator cell counts (prior to mixing with prey) would better fit in a follow-up study. However, please note that all experiments comparing predation on different prey strains were conducted in parallel and using the same predator inoculum.

Regarding the *popZ* mutant, assessing the predator:prey ratio would be even more challenging due to its larger and broad cell size distribution compared to wild-type strains. Furthermore, the impact of increased cell surface on the probability of co-infection would introduce another bias, complicating unambiguous interpretation related to co-infection.

c. L200 and associated figure. Can the authors rule out protein unfolding due to environmental changes such as pH change, as opposed to digestion of prey protein? To make this conclusion, some other assay will also have to be used (such as mass spec measurement of total protein content of prey population during predation). The section could be rewritten to include other possibilities for the observation of mCherry signal loss.

Response 3. We agree with the reviewer that some environmental changes resulting from predatory attack may impact protein folding inside the prey. However, given the firm junction between prey and predator and the absence of immediate leakage of the protein content, we do not think that the prey cytosol is exposed to the external environment. More importantly, *B. exovorus* is known to possess 113 putative peptidases and lacks numerous amino acid synthesis pathways (Pasternak et al., 2014 PMID: 24088628), indicating reliance on an exogenous supply from prey and underscoring the essential role of proteolytic activity. We have **added this information** in the revised discussion (Lines 373-375 in the tracked changes file). Furthermore, we have also included a control example in **revised Supplementary Fig. 3b**, demonstrating that the observed signal loss is not attributable to photobleaching (see also response 25 to Reviewer 4).

d. Discussion: could the authors perhaps speculate on what constitutes the attachment junction? Are there hints of some protein complexes that play a role in the same, based on the genes encoded by *B. exovorus*?

Response 4. The molecular nature of the attachment junction is an exciting question that we hope addressing in the future. Our intuitive guess is that it could involve *B. exovorus* outer membrane proteins and/or lipoproteins, potentially organized in a large assembly capable of connecting to the prey outer membrane. However, we currently lack specific indications regarding which protein might be included in this junction, and we have no reason to rule out other possibilities. If the reviewer agrees, we prefer to avoid unsupported speculation in our discussion, and keep this reflection for future studies.

Minor comments:

a. Figure legends: The number of experimental repeats and total number of cells imaged should be provided in all legends.

Response 5. We have **added this information** where appropriate.

b. L57: against protozoa, nematodes and bacteriophages

Response 6. Corrected, thanks.

c. L109: could the buffer conditions be included in the main text, or the sentence rewritten for clarity
Response 7. The buffer conditions are detailed in the **Methods** section (Lines 415-418 in the in the tracked changes file). We **rephrased** this sentence of the main text by removing “in a buffer”, which was not informative at this place, and replacing “incubation” by “co-incubation” (Line 144 in the tracked changes file).

d. In the example provided to suggest that *B. exovorus* can attach to the stalk as well (Fig. S1B), it appears that the predator is attached to both the stalk and the cell body. To clearly support ‘stalk-only’ attachment, another example should be included.

Response 8. Supplementary Movie 3 of the cell shown in the bottom right image of revised Fig. S1d clearly shows the stalk attachment. We have indicated the image better in the **revised text** (Line 191 in the tracked changes file) **and legend**. As suggested, we have **added time-lapse images** of an additional region of interest showing that *B. exovorus* can attach to the prey’s stalk in the **revised Supplementary Fig. 1e**.

e. L132: how was attachment frequency calculated?

Response 9. The frequency of attachment corresponds to the fraction of *C. crescentus* cells in direct contact with a predator cell, from images taken 15 minutes after perfusing the predator suspension in the microfluidic chamber containing the preys. We have **added the missing information** in the legend of Fig. 2c.

f. L158: Fig. 3a is referred to support formation of only two progenies. However, the highlighted cell appears to have three progenies that become two at T286min. Perhaps another example should be provided.

Response 10. Thank you for noticing this issue. We mistakenly referred to Fig. 3a to support formation of two progenies instead of Supplementary Fig. 2a. We have **corrected** the reference in the text (Line 213 in the tracked changes file) and removed the cyan arrowhead in Fig. 3a, which was confusing as it showed the first release of one out of three progenies. Note that Supplementary Fig. 2a (using the WT prey), 2e (using the Δ rsaA prey) and Supplementary Fig. 3a (using the WT prey) all include additional examples of 3 progenies and 2 progenies.

g. L247: such ‘a’ complex

Response 11. Corrected.

h. L454: predation ‘of’

Response 12. Corrected.

i. Fig. S1 legend: depicted ‘in’ Fig. 1a

Response 13. Corrected.

j. Fig. S2 legend: d-e reference for rsaA deletion and popZ deletion needs to be corrected

Response 14. Done. Thank you.

Reviewer #2

The manuscript describes the lifecycle of *B. exovorus*, revealing predation through an S layer and a novel cell division system. The quality of the microscopy is superb and the data presented supports the conclusions. I have only a few minor points:

Figure 1b The curves of predation appear to be different for the different prey strains with the mutant OD beginning to reduce before the wild-type, resulting in a different S value although the rmax rates look similar after this delay. Is this delay due to the presence of the S-layer? The S and rmax values for each biological repeat should be presented and statistical analysis applied to these values. There may well be a statistically significant difference in S that should be commented upon.

Response 15. We conducted the suggested **statistical analyses** comparing our four biological replicates and found that the r_{max} and s values were not statistically different (**revised Supplementary Fig. 1a and 1b**). However, there remains a consistent trend where the predator reaches its r_{max} slightly earlier when using the *rsaA* mutant as prey (Supplementary Fig. 1b). This small difference observed only at the population level could be related to the presence of the S-layer, but we cannot draw conclusions regarding the mechanism. Predator attachment (Fig. 2c) and predation time (from attachment to escape from the mother cell; revised Supplementary Fig. 2f) are similar when the predator utilizes the wild-type strain or the *rsaA* mutant as prey. See also Response 1 to Reviewer #1.

Lines 185-187 There are statistically more occurrences of 4 progeny when preying upon the larger cells and whilst rare, this does suggest that progeny numbers are affected by larger size, albeit not to the extent shown for *B. bacteriovorus*. Given that the relationship between numbers of progeny, progeny size and prey size don't equate in the same way as for *B. bacteriovorus*, was there any other differences when preying upon larger prey, for example numbers of storage granules?

Response 16. These are indeed particularly rare events (4 cells out of 201). The statistical significance most likely results from the comparison to 0 cases observed on WT preys. It does not rule out occurrence of >3 progenies on WT preys at low frequencies, which might be missed in our quantifications (the corkscrew-shaped elongation of predators and the fast sequential release of progenies sometimes prevent fully conclusive progeny counts). We think that if a larger prey size affected progeny numbers, we would have observed more cases of >3 progenies (and possibly >4 progenies). Besides, despite the important cell-to-cell variability in cell length of the *popZ* mutant, **our observations did not suggest any single-cell correlation between prey size and the few 4-progeny instances**. For all these reasons, we decided to use caution here by not concluding an a biologically relevant impact of prey size on progeny numbers.

We did not find any other notable differences when comparing the growth of *B. exovorus* using a smaller or larger prey (e.g., **revised Supplementary Fig.2f**). The storage granules are still present in the ghost cells upon predation (as seen in cryo-EM), suggesting that their content is not used for *B. exovorus* growth, and we did not notice any trend between the presence/number of granules and predator progeny counts.

Line 323 should this read "a different type" or "different types"?

Response 17. Different types. This is now **corrected**.

Line 372 How many predators were added? Pfu?

Response 18. As explained in Response 2 to Reviewer 1, we do not currently have a reliable quantitative method to precisely count predator cells. Plaques do not form as clearly for *B. exovorus* as for *B. bacteriovorus*. While we are implementing methods to address this technical limitation, we use procedures that are as standardized as possible to cultivate *B. exovorus*, prepare prey suspensions, and perform killing curves. Comparisons between different preys within one assay are always done with the same predator inoculum, the volume of which was empirically determined during our optimization of the killing assay.

Lines 431-36 How was normality tested for?

Response 19. We used the Shapiro test to assess the normality of our data. This information was added in the **revised "Statistical analyses" section** (Line 538 in the tracked changes file).

Line 494 representative

Response 20. Corrected. Thank you.

Reviewer #3

The work presented in the manuscript by Santin et al is of interest to a wide audience. The identification of key components of the cell cycle of *B. exovorus* during predation is exciting and original work. Bacteria predator-prey interactions remain poorly understood and this work provides insights about the diversity of strategies/lifecycles of the predator bacteria. Most of the conclusions have data to support them with well

thought-out methodologies and experimental plans. I only have a few comments. Overall, I enjoyed reading this manuscript.

Introduction section: More background information about *B. exovorus* is necessary. For instance, does *B.e.* have an S-layer? Figure 5 suggest that *B.e.* has no S-layer. Is that correct? Is *B.e.* an obligate epibiotic or can they grow in the absence of a bacterium prey?

Response 21. Little is known about this bacterium since its initial characterization in 2013 (Koval et al., 2013). Our study is the first to provide live imaging and measurements of biological parameters. The reviewer is right, our cryo-EM analyses did not reveal the presence of an S-layer on *B. exovorus* (Fig. 5). We have **added this information** in the revised figure legend. Consistently, no *rsaA* gene homolog was found in the genome. *B. exovorus* is indeed an obligate predator that cannot grow in the absence of prey, as previously shown (Koval et al., 2013). We **added this information** in the revised introduction (Line 99 in the tracked changes file).

Line 161 – 164. I am not convinced the data supports the conclusion that “*B. exovorus* initiates cell constriction during growth (Fig. 3a)...” The data on Figure 3a are only a couple sets of cells that seem to constrict but the figures are too small to really deduce anything. This potential constriction during growth should be quantified. The delta-*popZ* filamentous cells could facilitate the quantification of this observation. On Figure 3C, the sequential division in the pre-divisional cells in the cryo-EM images is clear. However, it is possible based on the size of those cells that they were done growing and only then initiated division sequentially.

Response 22. Supplementary Movie 2 shows more clearly that cells keep elongating when constriction is already visible (see for instance the cell on the left side). We have **added the reference to this movie** in the revised text (Line 217 in the tracked changes file). We have also **added a zoomed region of interest in the revised Supplementary Fig. 2b** to better illustrate this aspect.

It would indeed be ideal to quantify the growth and division patterns of *B. exovorus*. However, this is currently technically challenging. Indeed, we observed that *B. exovorus* coils as it grows, and our microfluidic system allows them to do so in “3D” (since they are not squeezed between a pad and coverslip – we have added this note in the legend of the revised Supplementary Fig. 3b). This hinders the precise measurement of morphological parameters such as cell length. Our numerous attempts to grow *B. exovorus* on agarose pads after mixing with prey have been unsuccessful so far. We do not think that using the *popZ* deletion strain would facilitate measurements of growth and constriction since we did not observe a correlation between prey size and progeny numbers (see also Response 16 to Reviewer #2).

The data collected from time lapses could be used to deduce a couple other important parameters. For instance, does the prey size have any correlation with the number of *B.e.* that adhere? Can multiple *B.e.* prey on a single *Caulobacter* bacterium? Is there a fixed time of predation? In bigger cells (delta-*popZ*), was the predation time longer than in smaller wildtype *Caulobacter* cells? A constant time of preying would support the comments given on lines 283 – 286.

Response 23. Yes, multiple *B. exovorus* can attach, grow, and release progeny on a single *C. crescentus* cell. Although co-infection appears to be primarily dictated by the initial quantity of injected predators, the probability of co-infection increases significantly when prey cells are very elongated, such as the *popZ* mutant (due to the increased exposed surface area). However, this variable is empirically controlled in our experiments by using a low amount of predators to prioritize single-cell analysis (see also Response 2 to Reviewer 1).

As suggested, we have **measured the predation time** of *B. exovorus* (defined here as the time interval between attachment to the prey and the escape of the attached predator after releasing the progeny) in the presence of each prey strain, for predators that produce 2, 3, or 4 progenies (see **revised Supplementary Fig. 2f**). We did not find any significant differences of predation time between wild-type and *popZ* prey. These data indeed support our conclusion that the growth of *B. exovorus* is less adaptable than that of *B. bacteriovorus* (Santin et al., 2023). Thank you for the suggestion.

The results section or at least the discussion section would benefit from a more thorough analyses of the bacterial species that are presented on Figure 1BC. Based on the outer membrane/cell envelope genes found between susceptible and resistant species, is there anything that could explain the differences in susceptibility to B.e.? Based on bioinformatic analyses, is there anything that the authors could predict would cause the susceptibility? This information could broaden the impact of the findings and provide a big picture to the discussion section.

Response 24. The characterization of resistance factors to *B. exovorus* will constitute a separate study. We aim to conduct a systematic investigation on a larger sample of prey species. At present, we refrain from speculating on the nature of the elements conferring resistance. However, we anticipate that one or more components of the outer membrane (OM) and peptidoglycan (PG) may be involved. Please note that we followed advises received independently of this revision to mention the order and not just the families in our phylogenetic tree, which better emphasizes the proposed evolutionary diversification resulting in predation-sensitive or resistant species (**revised Fig. 1b and related text**, Lines 161-163 in the tracked changes file).

Reviewer #4

Santin et al. investigate the life cycle of the predatory bacterium *Bdellovibrio exovorus* and the fate of its host bacterium *Caulobacter crescentus* during the infection cycle using optical microscopy coupled with cryo electron-microscopy (cryo-EM). The authors show that *B. exovorus* can grow on a restricted range of alpha-proteobacteria and even infect a *C. crescentus* wild-type strain expressing an S-layer, challenging a previous statement suggesting that the S-layer protects against predatory bacteria. Life-cell imaging data demonstrate that *B. exovorus* preferably reproduce triplet progenies independent of the host cell size. Overall, the experiments are well designed, the data is professionally documented, and the manuscript is well-written. I have a few suggestions which should further strengthen the manuscript.

The authors use a cytosolic mCherry *C. crescentus* strain to investigate the fate of the host cytoplasm. Loss of fluorescent intensity signal is interpreted as digestion/degradation of the host cytoplasm through *B. exovorus*. The infected host cells are imaged for 2 hours at every 5 intervals and then in 2 minutes intervals (Methods, lines 394-395). Photobleaching of cytoplasmic mCherry might partially contribute to the loss of fluorescent intensity signal over time. Therefore, I think it would be useful to image uninfected control host *C. crescentus* cells under identically imaging conditions and compare the fluorescent signal over time to the existing data of infected *C. crescentus* cells.

Response 25. We agree with the reviewer that photobleaching could have potentially explained the loss of mCherry signal over time. However, this decrease of fluorescence signal was not observed in uninfected cells present on the same field of view. A representative example has been added as a control to the **revised Supplementary Fig. 3**. Thank you for the suggestion.

Lines 299-309: The authors suggests that the host cytoplasm is actively been degraded by secreted protease, nucleases, lipases from *B. exovorus*. Could the author comment or discuss if the *B. exovorus* genome encodes these effectors?

Response 26. According to a previous study, *B. exovorus* encodes 113 putative peptidases and 17 ribonucleases (Pasternak et al., 2014), in addition to 15 lipases that we have detected ourselves. This abundance of degradation enzymes is comparable to that observed in *B. bacteriovorus* (Rendulic et al., 2004; Pasternak et al., 2014). We have **added this information** in the revised discussion (Lines 373-375 in the tracked changes file).

Lines 311-320: The authors suggests that the outer membrane of *B. exovorus* and its host *C. crescentus* do

not fuse during the infection cycle based on lack of transfer of the S-layer from the host to the epibiont. The S-layer is known to bind to the O-antigen of the host lipopolysaccharide and neighbouring S-layer subunits. The lack of S-layer transfer to the epibiont might be due to strong interaction of RsaA subunits within S-layer lattice which prevents diffusion. Given the fast attachment of *B. exovorus* to host cells within 15 minutes (line 133), should it not be possible to pulse label host *C. crescentus* time lapse-microscopy after mixing pulse-labelled host cells with the epibiont and investigating if the fluorescent signal remains

Response 27. We have conducted the suggested experiment by labeling prey cells with FM4-64 before co-incubation with the predator (see **Reviewer Figure 1** below). Strikingly, we observed that the FM4-64 signal rapidly disappears after attachment of *B. exovorus*. This effect is not due to photobleaching, as the signal remains constant over time for uninfected cells. Although a reorganization and/or partial destruction (e.g., of the inner leaflet) of the prey outer membrane resulting from the predatory attack might explain these results, we currently don't have an explanation for this phenomenon. No transfer of fluorescence from the prey to the predator was observed, but it is presently difficult to unambiguously interpret this result given the total disappearance of the FM4-64 signal early upon infection. While this is worth future investigation, we would prefer not to include it in the revised manuscript.

Reviewer Figure 1. The prey outer membrane is likely not transferred to the predator during growth. Representative time-lapse phase contrast microscopy images of *B. exovorus* growing onto the wild-type *C. crescentus* prey incubated with FM4-64 prior to infection. White arrowheads indicate the position of *B. exovorus* cells on a *C. crescentus* cell. The FM4-64 fluorescent signal quickly decreases upon *B. exovorus* attachment compared to uninfected prey cells. No transfer of the FM4-64 signal was observed during predator growth. Scale bar, 2 μ m. Related to Fig. 5.

Regarding the point raised above, I think it would be useful to obtain a cryo electron tomogram of the "feeding pole" to show the discontinuity between the outer membranes of the epibiont and the host cell.

Response 28. We agree with the reviewer that cryo-electron tomography would be highly informative by providing insights into epibiont-prey contact with greater details. However, we feel that such highly technical experiments would better fit in a future study focused on elucidating the molecular nature of the feeding junction. Please see also Response 4 to Reviewer 1.

Minor comments:

Lines 112,343; Figure 1b,c: subvibriodes-> subvibrioides missing "i".
<https://www.ncbi.nlm.nih.gov/datasets/taxonomy/74313/>

Response 29. Corrected, thank you.

Line 133: "WT" -> wild-type (see line 135).

Response 30. Done.

Line 164: Cryo-EM -> The abbreviation has not yet been specified.

Response 31. Done.

Line 243: (n = 48 cells) -> n should be italicised.

Response 32. Done.

Line 367: MUSCLE -> Reference/Citation missing.

Response 33. We have added the missing reference.

Line 425: GitHub -> Please provide the link in the method section or data/software accessibility.

Response 34. Done

Lines 454/458: "α"-proteobacterial. The main text refers to "alpha"-proteobacteria (see line 88, 111, 120, ...). Use either α or alpha throughout the manuscript.

Response 35. Done.

Line 481: (left) Scale bar, 0.5 μm -> Full stop missing between "(left)" and "Scale bar".

Response 36. Done.

Line 509: Fiji -> The method section only refers to ImageJ (version 1.53t). Reference/Citation missing for Fiji or ImageJ.

Response 37. Thank you for pointing this out. We have added the missing reference for Fiji in a new paragraph on image processing the methods (Lines 530-533 in the tracked changes file).

Line 590: Cryo-EM imaging of predator:prey junction using the ΔrsaA *C. crescentus* -> *C. crescentus* should be italicised.

Response 38. Done.

Reviewer #1 (Remarks to the Author):

All my previous comments have been satisfactorily addressed.

I only have one minor follow-up comment: I request authors to include a sentence in the results or discussion about the limitation with regards to calculating the exact predator load in the current experimental setup.

Reviewer #2 (Remarks to the Author):

The revised manuscript addresses the concerns of the reviewers and has improved it.

Reviewer #3 (Remarks to the Author):

My concerns have been addressed in the revisions

Reviewer #4 (Remarks to the Author):

I would like to thank the authors for revising their manuscript. The new co-incubation data does seem to indicate no fluorescent is transferred from the host cells to the predators despite the loss of signal of infected cells. While I still believe that cryo electron tomography data of the feeding pole would be informative to elucidate the feeding mechanism, I agree with the response of the authors that this would require a detailed future study. Therefore, I would like to congratulate the authors for a nicely revised manuscript.

Response to the reviewers – Santin, Sogues, et al.

REVIEWERS' COMMENTS

Reviewer #1 (Remarks to the Author):

All my previous comments have been satisfactorily addressed.

I only have one minor follow-up comment: I request authors to include a sentence in the results or discussion about the limitation with regards to calculating the exact predator load in the current experimental setup.

We have added this information in the Methods section.

Reviewer #2 (Remarks to the Author):

The revised manuscript addresses the concerns of the reviewers and has improved it.

Reviewer #3 (Remarks to the Author):

My concerns have been addressed in the revisions

Reviewer #4 (Remarks to the Author):

I would like to thank the authors for revising their manuscript. The new co-incubation data does seem to indicate no fluorescent is transferred from the host cells to the predators despite the loss of signal of infected cells. While I still believe that cryo electron tomography data of the feeding pole would be informative to elucidate the feeding mechanism, I agree with the response of the authors that this would require a detailed future study. Therefore, I would like to congratulate the authors for a nicely revised manuscript.

We would like to thank all reviewers again for their time and their highly professional input that allowed us to improve our manuscript. We are delighted to see their positive comments about our revision.